# W-WaveNet: A multi-site water quality prediction model incorporating adaptive graph convolution and CNN-LSTM

**Shaojun Yang** ⓘ *◉, **Shangping Zhong** *◉, **Kaizhi Chen** ⓘ◉

College of Computer and Data Science, Fuzhou University, Fujian, 350108, China

◉ These authors contributed equally to this work.
* shaojunyoung@gmail.com (SY); spzhong@fzu.edu.cn (SZ)

**Data Availability Statement:** All data are available from https://github.com/LeonardYoung/WaveNet_LM/tree/main/data.

**Funding:** The author(s) received no specific funding for this work.

## Abstract

Water quality prediction is of great significance in pollution control, prevention, and management. Deep learning models have been applied to water quality prediction in many recent studies. However, most existing deep learning models for water quality prediction are used for single-site data, only considering the time dependency of water quality data and ignoring the spatial correlation among multi-sites. This research defines and analyzes the non-aligned spatial correlations that exist in multi-site water quality data. Then deploy spatial-temporal graph convolution to process water quality data, which takes into account both the temporal and spatial correlation of multi-site water quality data. A multi-site water pollution prediction method called W-WaveNet is proposed that integrates adaptive graph convolution and Convolutional Neural Network, Long Short-Term Memory (CNN-LSTM). It integrates temporal and spatial models by interleaved stacking. Theoretical analysis shows that the method can deal with non-aligned spatial correlations in different time spans, which is suitable for water quality data processing. The model validates water quality data generated on two real river sections that have multiple sites. The experimental results were compared with the results of Support Vector Regression, CNN-LSTM, and Spatial-Temporal Graph Convolutional Networks (STGCN). It shows that when W-WaveNet predicts water quality over two river sections, the average Mean Absolute Error is 0.264, which is 45.2% lower than the commonly used CNN-LSTM model and 23.8% lower than the STGCN. The comparison experiments also demonstrate that W-WaveNet has a more stable performance in predicting longer sequences.

## Introduction

Water resources are the most important material basis for human survival. In recent years, pollutants from industrialization, urbanization, and agriculture have entered the river, deteriorating its quality. Organic pollutants (pesticides, pesticides, phenols, hydrocarbons, and so on) as well as heavy metals (mercury, cadmium, copper, arsenic, lead, and so on) and microbiological pathogens are all discharged into the body of water [1]. These water pollutants have a

**Competing interests:** The authors have declared that no competing interests exist.

significant negative impact on human activities as well as the environment. As a result, it is critical to monitor and warn about water quality [2].

The mechanism water quality model and the non-mechanism water quality model are two types of water quality prediction models [3]. The mechanism water quality prediction model, for example, uses the pollution source as its major body and relies on a water quality physical model to mimic the diffusion process of water pollution factors in order to achieve the forecast goal [4]. This mechanical method, on the other hand, necessitates professionals adjusting parameters according to water conditions, which has weak generalization capacity and is influenced by a complex and variable mathematical process.

A data-driven method is a non-mechanistic water quality model that uses the receiving body of water as the main body. It has been widely researched [5] in recent years, particularly as machine learning has grown in popularity. In particular, the model based on deep learning has made important achievements in water quality prediction. Deep learning algorithms fit the nonlinear and unstable water pollution time series data effectively [6].

Deep learning [7] is a research field in machine learning. It has become a prominent method for analyzing water quality due to its good adaptability to uncertainties and non-linear circumstance [8,9]. With the development of the Internet of Things, more and more water quality pollution data are transmitted to the cloud platform through embedded devices in an automated process [10–14]. The continuous running of multi-sites and sampling equipment in the water quality monitoring system causes the data to grow over time. Meanwhile, the water pollution data itself has the characteristics of non-linearity, non-stationarity, and fuzziness [15], In addition, deep learning models have the following benefits [16]: (1) simplify system complexity to facilitate understanding and use, (2) predict target values when site visits are problematic, (3) reduce time and cost, and (3) facilitate prediction at all phases of the system. Many studies integrate deep learning methods with other machine learning methods that. Recently, more researchers are integrating deep learning methods with other machine learning methods and have achieved better results than a single deep learning model. Based on the LSTM model, Jia et al. [17] combined Back Propagation Neural Network (BPNN) to form the LSTM-BP model. Barzegar et al. [16] combined CNN and LSTM to propose a CNN-LSTM prediction model. This research implemented SVR and a decision tree and compared them to the CNN-LSTM model to compare with a shallow machine learning model. Jian Zhou et al. [18] combined LSTM with the grey correlation method. Firstly, considering the multiple correlations of water pollution information, a feature selection algorithm for water pollution information based on similarity and proximity is proposed. Secondly, considering the time series of water pollution information, a water pollution prediction model based on LSTM is established. The feature obtained by Improved Grey Relational Analysis (IGRA) serves as the model's input. Finally, the approach is tested on two real-world data sets for water pollution in Taihu Lake and Victoria Bay. The results of the experiments reveal that this strategy can fully utilize multivariate correlation and time series data on water pollution.

It can be concluded that the present deep learning-based water pollution analysis model focuses mostly on integrating the LSTM model with other machine learning methods. These models' drawbacks include (1) difficult parallel computation, long training and evaluation time, (2) Possible problems of gradient explosion and gradient disappearance, (3) ignore the relationships among multiple water quality monitoring sites. In reality, multiple water quality monitoring sites are often placed on a river section, and there is a strong correlation among these sites, which we call spatial correlation. An ideal model should be able to estimate how downstream data would be affected when there is abrupt contamination in the upstream. This can only be done with models that account for spatial correlations. Spatial correlation is as complex as temporal correlation. The flowing velocity, surroundings, and season can all have

an impact on the spatial dependency of water quality data, making it nonlinear and unstable. It has been demonstrated that deep learning is effective at handling this kind of data. In this paper, a model using graph space-time is proposed for the first time to deal with spatial dependencies in water quality data.

In order to deal with both time dependence and spatial dependence, this paper chooses graph spatio-temporal convolution to model water quality pollution data. Spatial-temporal graph convolution is a type of Graphic Neural Network (GNN) that is used to solve problems involving dynamic graphs. It forms multiple water quality monitoring sites into a graph and uses the relationships among nodes in the graph to represent the spatial relationships among water quality monitoring sites. Spatial-temporal graph convolution employs graph convolution to capture spatial correlations while time correlations are handled in a variety of ways. Based on the way of dealing with time correlation, the spatial-temporal convolution can be split into two directions [19]: RNN-based and CNN-based. Many types of studies are currently based on RNN. For forecasting dynamic MINIST data and fitting natural language data, Seo et al. [20] combined graph convolution networks with LSTM to construct model. The DCRNN model proposed by Li et al. [21] incorporates the graph convolution layer into the Gate Recurrent Unit (GRU) network and is used for traffic forecasting problems. For graph representation learning, Taheri et al. [22] developed a spatial-temporal graph convolution model that includes self-encoders, GGNNs (Gated Graph Neural Networks), and LSTM. A group of wild baboons' GPS tracking data was used to test the model.

Both RNN-based and CNN-based spatial-temporal graph convolution models have advantages and disadvantages. Although the former has inherent advantages when working with time-series data, iteration propagation might result in gradient explosion or disappearance. CNN-based models, on the other hand, have the advantages of parallel computation, a constant gradient, and minimal memory usage. Yu et al. [23] proposed the STGCN model, which combines the one-dimensional convolution layer with ChebyNet convolution. By stacking a gated 1D convolution layer, a graph convolution layer, and another gated 1D convolution layer in order, it creates a spatial-temporal block. ASTGCN was proposed by Guo et al. [24] to tackle time dependence using CNN. In addition, both temporal and spatial convolutions are enhanced by attention processes. In terms of traffic forecasting, the model performs well.

In summary, CNN-LSTM hybrid models have been widely used and achieved excellent performance in the field of water quality analysis in recent years, but these models do not consider the spatial correlation in the case of multi-site. The spatio-temporal graph convolution model, which considers both temporal and spatial correlations, has achieved great results in many fields. Therefore, in this study, the CNN-LSTM model is fused into the spatio-temporal graph model and applied to the field of multi-site water quality prediction.

Specifically, this study uses an adaptive graph convolution model to automatically learn the relationships among sites. To create a CNN-LSTM network, the TCN network from the Wave-Net [25] model is extracted and combined with the LSTM network. Finally, the adaptive graph convolutional network is fused with the CNN-LSTM network. Due to the applicability of the model to water quality data, we call it W-WaveNet, where the first W is an abbreviation for water.

The innovations of this study are as follows.

1. This paper defines and analyzes the non-aligned spatial correlations existing in multi-site water quality data before deducing the input-output interdependence of the spatio-temporal blocks described in this paper using formulae of one-dimensional and graph convolution. Theoretical analysis demonstrates that the spatiotemporal block can handle non-aligned spatial dependencies in water quality data and can be used for time spans of various lengths.

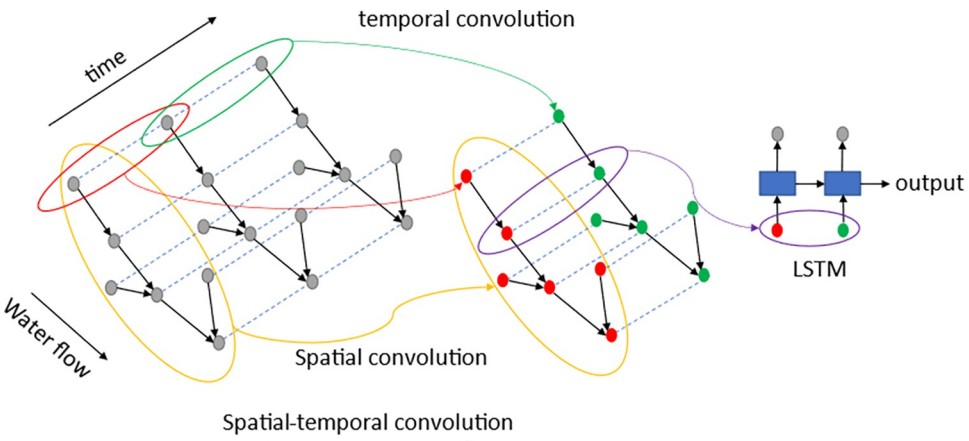

**Fig 1. Schematic diagram of the fusion model this model integrates WaveNet network, LSTM network and GCN network.** The data are first processed using convolution in the temporal dimension, followed by a spatial convolutional network in the spatial dimension, and finally reinforced by an LSTM network to correlate the front-to-back dependencies of the data.

2. In this paper, the three networks are fused together by interleaved stacking of multiple WaveNet and adaptive graph convolutional networks followed by connecting LSTM. As shown in Fig 1, the WaveNet network is used to extract local features, the LSTM network is used to model data feature dependencies, and the adaptive graph convolution network is used to model spatial association relationships.

Compared with classical water quality pollution prediction models, the method proposed in this paper additionally considers complex spatial correlations. The major advantage of our suggested model over previous models is that it can manage spatial dependencies among numerous nearby sites and complex spatial relationships. Compared with the similar work of Graph WaveNet [26] which used only for traffic speed prediction, the method proposed in this paper (a) improves the adaptive graph convolution network; (b) adds the LSTM network; (c) improves the spatio-temporal network fusion strategy; and (d) adds the residual connectivity.

The source code and dataset of the model can be accessed from https://github.com/LeonardYoung/WaveNet_LM.

The rest of the article is organized as follows. The second chapter introduces the model for this paper. The third chapter introduces the dataset and goes into detail on the experimental process. Finally, the fourth chapter concludes.

## Materials and methods

### Dataset describtion

The data used in this paper comes from two sections of a river basin in Fujian. With the permission of the ecological and environmental protection government department, the data is automatically collected by the water quality online monitoring equipment and then uploaded to the cloud platform through the network. The data is now publicly available in the GitHub repository. For convenience, they are hereinafter referred to as section A and section B, respectively. These two sections are located on different tributaries of the same river. There are 10 monitoring stations in Section A, some of which are connected by rivers, and others are connected by underground pipe networks, as shown in Fig 2.

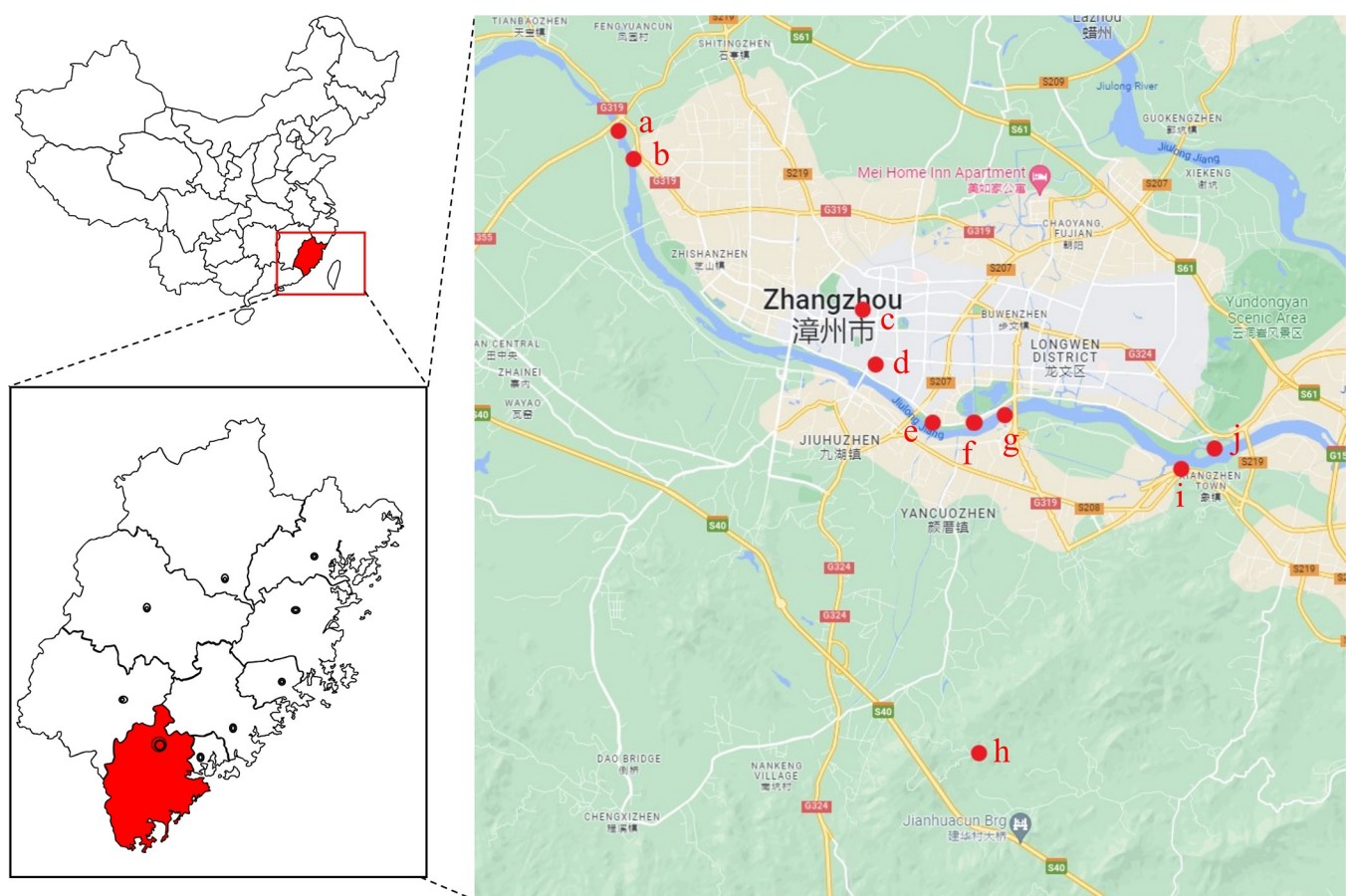

**Fig 2. The geographical position and site distribution of section A.**

Data loss is unavoidable. In order to select a time period with high data quality for research, the monthly loss rate of data is counted. The hourly data from January 1, 2020, to October 31, 2020 has been selected. The total number of data points is about 24000, and each data point has six water quality factors: pH value, total nitrogen (TN), total phosphorus (TP), ammonia nitrogen ($NH_3$), dissolved oxygen (DO), and permanganate index ($COD_{mn}$). The data interval is 4 hours. The data statistics are shown in Table 1. The units of all pollution factors are mg/L except pH.

The same method is adopted to filter the data in section B. Finally, the data from October 1, 2019 to September 30, 2021 has been selected. The total amount of data is about 30.000. The data interval is also 4 hours. Data statistics are shown in Table 2.

**Table 1. Data statistics of each water pollution factor in section A.**

| pollution factors | min | max | mean | std | skewness | kurtosis |
|---|---|---|---|---|---|---|
| pH | 4.111 | 9.023 | 6.55 | 0.957 | -0.164 | -0.18 |
| TN | 0.54 | 13.007 | 6.471 | 2.361 | 0.519 | -0.144 |
| TP | 0.045 | 1.172 | 0.447 | 0.271 | 0.815 | -0.241 |
| $NH_3$ | 0.003 | 5.963 | 1.752 | 1.48 | 0.949 | -0.193 |
| DO | 0.005 | 11.681 | 4.647 | 2.437 | -0.209 | -0.968 |
| $COD_{mn}$ | 0.123 | 11.167 | 5.092 | 2.407 | 0.863 | -0.263 |

**Table 2. Data statistics of each water pollution factor in section B.**

| pollution factors | min | max | mean | std | skew | kurt |
|---|---|---|---|---|---|---|
| pH | 5.105 | 9.482 | 7.271 | 0.832 | -0.085 | 0.078 |
| TN | 0.06 | 15.33 | 6.417 | 3.645 | 1.01 | 0.11 |
| TP | 0.002 | 1.1 | 0.383 | 0.261 | 0.858 | -0.088 |
| NH$_3$ | 0.002 | 7.93 | 2.154 | 2.053 | 1.087 | 0.319 |
| DO | 0.002 | 12.222 | 5.451 | 2.408 | -0.337 | -0.482 |
| COD$_{mn}$ | 0.1 | 11.05 | 5.386 | 2.139 | 0.543 | -0.109 |

By comparing the data in Tables 1 and 2, it can be concluded that there are significant differences in the pollution of the two sections. For example, for pH value, the mean value of section B is 11% more than that of section A, but the standard deviation is smaller. The pH skewness of the two sections is less, but the pH kurtosis is very different. There are differences to varying degrees among the six pollution factors, among which there are great differences in pH, NH$_3$ and DO.

## Problem definition

Given a graph $G = (V,E)$, where $V$ and $E$ are the sets of nodes and edges, respectively. At the moment $t$, the time series data on the nodes are represented as $X^t \in \mathbb{R}^{n \times f}$, where $n$ is the number of graph nodes and $f$ is the length of data features. Let $S$ represent the input sequence length and $P$ denote the prediction step, then the problem of this paper is to find the mapping relation $F$ in Eq (2-1).

$$X^{(t-s):t}, G \xrightarrow{F} X^{(t+1):(t+P)} \quad (2-1)$$

Considering the case of a single site, the dependencies of the data can be expressed using a Markov chain.

$$X^{t-1} \xrightarrow{P(X^t | X^{t-1})} X^t \quad (2-2)$$

We extend this to the case of multiple sites. The features among the nodes are correlated, but it is worth noting that the correlations between node features are not aligned in the time dimension. Let the features at moment $t$ be $X^t \in \mathbb{R}^{n \times f}$. Then $X_i^{(t)}$ denotes the feature of the ith node at moment $t$. The association relationship can be represented as

$$X_{0:i}^{(t-k):t} \rightarrow X_i^{(t)} \quad (2-3)$$

where $k$ is a positive integer greater than or equal to 0. This means that the features of node $i$ at moment $t$ and the features of nodes before $i$ from moment $t$-$k$ to moment $t$ may be correlated. In the water quality prediction scenario, this characteristic is actually the spatial correlation mentioned above, which will be described in depth in Chapter 3.1. Direct graph convolution operation on $X^{(t)}$ ignores the non-alignment of this spatial correlation and is undesirable. How to describe this correlation in a model is the difficulty of the problem.

## Relative models

**Adaptive graph convolution network.** Given the graph structure, the GCN (Graph Convolution Network) model is essentially an operation of aggregating graph neighborhood information. The first-order approximation of the ChebyNet spectral filter was proposed by Kipf [27]. It smooths the signal of nodes by aggregating and manipulating the neighborhood

information of nodes from a spatial perspective. The advantage of this method is that its filter is positioned in space as a composition layer, and it allows multi-dimensional input. GCN can be simply described by the Eq (2-4). Where $A \in \mathbb{R}^{n \times n}$ denotes the adjacency matrix, $n$ is the number of nodes in the graph, $W \in \mathbb{R}^{n \times n}$ is the model parameter, $X \in \mathbb{R}^{n \times f \times S}$ is the model input, and $S$ represents the input sequence length of water pollution data.

$$Z = AXW \qquad (2-4)$$

In this paper, the adjacency matrix $A \in \mathbb{R}^{n \times n}$ is defined in Eq (2-5).

$$A_{ij} = \begin{cases} 0 & e_{ij} \notin E \\ \dfrac{1}{d_{ij}} & e_{ij} \in E \end{cases} \qquad (2-5)$$

Where $d_{ij}$ is the distance among nodes. The larger the distance, the smaller the node association. However, in some application scenarios, it is often difficult to obtain the node distance $d$ for various reasons. Graph adaptation is a good solution in such cases. Graph adaptation refers to a method of learning a graph adjacency matrix from data, which has been widely used in various problem domains. This data-driven approach increases the flexibility of the graph construction model, making it more general and able to adapt to different data samples.

The adaptive model used in this paper refers to the graph adaptive module proposed by Shi et al. [28], which can be describe as Eq (2-6).

$$X_{out} = (A_\alpha + B_\alpha + C_\alpha) X_{in} W \qquad (2-6)$$

Where $X_{in}$, $X_{out}$ represents the input and output data respectively. $A_\alpha$, $B_\alpha$, $C_\alpha$ is the three parts of the adaptive module. $W$ is a graph convolution parameter. The overall architecture of the module is shown in Fig 3. Where the orange block represents learnable parameters.

The first part $A_\alpha$ represents the original adjacency matrix, and this original adjacency matrix usually does not describe the relationship between nodes well due to measurement errors, calculation formulas, etc. Conversely, the model does not require the high accuracy of $A_\alpha$. When the node relationship is difficult to measure, $A_\alpha$ can even be removed directly. In this case, the formula turns into:

$$X_{out} = (B_\alpha + C_\alpha) X_{in} W \qquad (2-7)$$

The second part $B_\alpha$ is a network parameter that is initialized randomly before training. $B_\alpha$ is completely learned from data, which means that it will not be limited to any restrictions and can generate connections between nodes. This feature enhances the flexibility of the adaptive module.

The third part $C_\alpha$ represents the embedded representation of nodes obtained through data learning. It can be described as Eq (2-8):

$$C_\alpha = SoftMax(ReLU(E_1 E_2^T)) \qquad (2-8)$$

Where $E_1$, $E_2$ is the randomly initialized matrix, which represents the node embedding of the source node and the target node, respectively. The spatial correlation weight between the source and destination nodes can be calculated by multiplying $E_1$ and $E_2$. Use the ReLU activation function to eliminate connections with low correlation. After that, the softmax function is employed to normalize the adaptive adjacency matrix. The adaptive graph convolution network is hereinafter referred to as AGCN network.

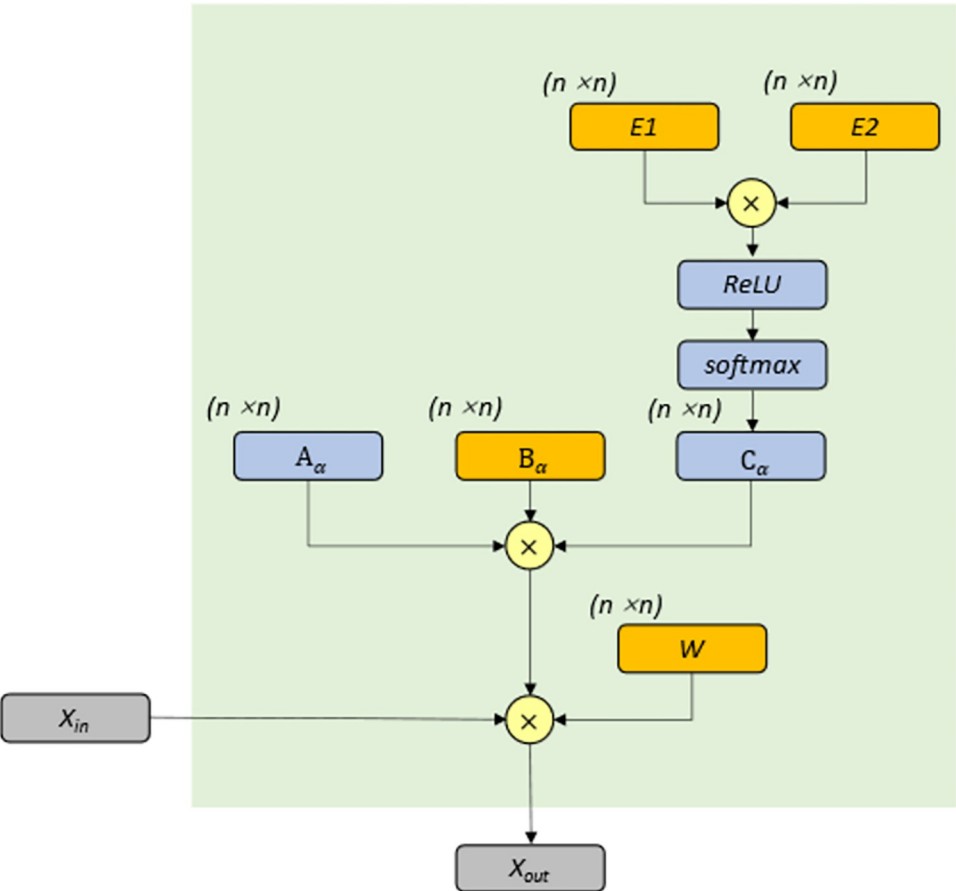

**Fig 3. Framework of the adaptive graph convolution.** It contains 3 components $A_\alpha$, $B_\alpha$, and $C_\alpha$. $A_\alpha$ is used to represent the geographical information of the nodes, $B_\alpha$ is a random initialization matrix to enhance the flexibility of the model, and $C_\alpha$ is used to represent the learnable node embeddings.

## WaveNet network

WaveNet is a deep neural network that is used to create raw audio waveforms, and it has recently been employed in several researchs for time series analysis with promising results. The WaveNet network is made up of several identical blocks, each of which has a dilated causal convolution [29], a gated convolutional network, and a residual connection [30]. Fig 4 depicts

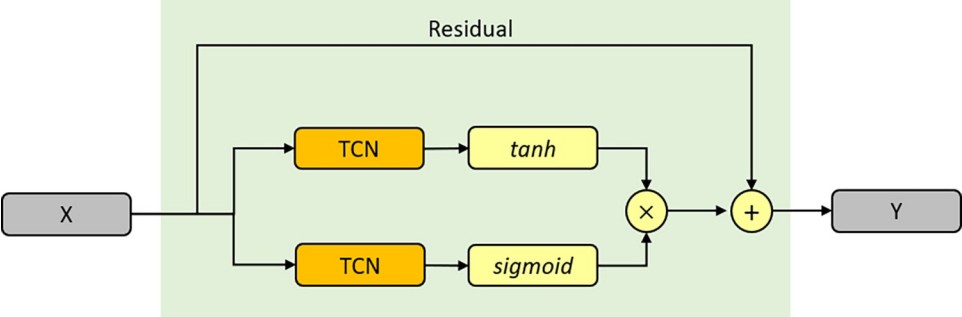

**Fig 4. Framework of the WaveNet block.** It employs dilated causal convolution as a foundation and includes a gated mechanism and a residual connection.

the structure. The dilated causal convolution is indicated by the TCN in the illustration. Unlike the RNN-based approach, dilated causal convolution can handle large time sequences in a non-recursive way, making parallel computation easier and alleviating the gradient explosion problem.

**CNN-LSTM hybrid model.** Convolutional neural networks leverage the concept of local correlation and data weight sharing to substantially reduce the number of parameters in the network, while also allowing for parallelism and quick training speeds. Although parallelizing RNN is complex, it offers a natural benefit when working with time series data. LSTM [31] is an upgraded RNN model that is presented to overcome the problems of gradient disappearance and gradient explosion during the training of long series data, and LSTM can have better performance in longer sequences. In recent years, some researches have combined CNN and LSTM to incorporate the benefits of both models [16]. The CNN component of the model extracts local trend elements as well as common features that exist at different intervals in the time series. Then, the LSTM learns sequential relationships by capturing short and long-term dependencies. Theory and practice suggest that combining CNN and LSTM models in a sensible way can produce better results than using them separately.

The hybrid model's construction is presented in Fig 5, with input data passing through an 1×1 convolution to increase dimensionality, then numerous CNN networks in sequence, whose results are plugged into the LSTM network, and lastly a linear network to output the results.

## Proposed methodology

**Spatio-temporal network fusion strategy.** In order to form a spatio-temporal graph convolution model, the temporal network needs to be fused with the spatial network. The fusion approach is shown in Fig 6. The data first pass through a one-dimensional convolution for the temporal dimension, and then the data of each output channel pass through a GCN network for the spatial dimension. This is repeated several times, and finally passes through an LSTM network.

Placing the GCN module behind the convolution allows the model to handle the non-aligned correlation of the data. After the data is subjected to the convolution operation, the

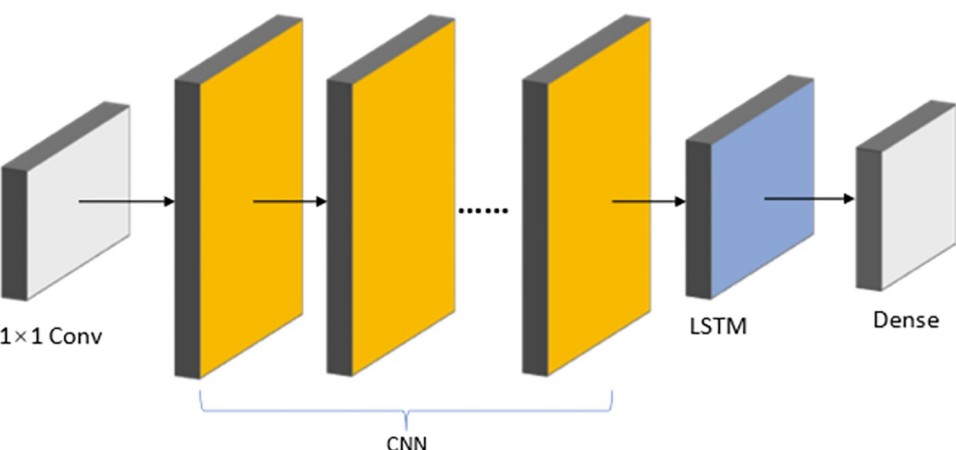

**Fig 5. Framework of the CNN-LSTM hybrid model.** It contains multiple convolution layers, which are finally output through an LSTM and a full connection layer.

**Fig 6. Schematic representation of spatio-temporal network fusion strategy.** The GCN network immediately follows the CNN to form a spatio-temporal processing module, which is stacked to handle spatial dependencies of different spans, and finally, the LSTM network outputs the results.

data of the $c$th output channel can be expressed as Eq (2-9)

$$\dot{X}_c = b_c + \sum_{i=0}^{c_{in}} w_c^i \otimes X_i \qquad (2-9)$$

where $c_{in}$ represents the number of input channels, and $b_c$ and $w_c^i$ are the convolution parameters of the cth output channel. The output data of this channel can be regarded as a time series, so the $t$-th data in the series is:

$$\dot{X}_c^t = b_c^t + \sum_{i=0}^{c_{in}} w_c^i \otimes X_i^{(t-k):t} \qquad (2-10)$$

Due to the property of convolution sharing weights, $w_c^i$ is unchanged. $k$ denotes the length of convolution kernel. The length of $X_i^{(t-k):t}$ in Eq (2-10) is equal to the length of convolution kernel. Without considering the computation process, the channel output data of the $p$th node can be expressed as Eq (2-11), which means that the $t$-moment data of the convolutional layer output result is determined by the $k$ numbers of input data before the moment t.

$$X_p^{(t-k):t} \xrightarrow{conv} \dot{X}_{c,p}^t \qquad (2-11)$$

The GCN operation is performed next, and according to Eq (2-4), the data need to be left multiplied by the adjacency matrix A. The result can be presented as Eq (2-12)

$$\dot{Y}_{c,q}^t = \sum_{p=0}^{n} a_{ip} \cdot \dot{X}_{c,p}^t \qquad (2-12)$$

where $n$ is the total number of nodes and $\dot{Y}_{c,q}^t$ denotes the feature of node $q$ in the $t$th dimension obtained by the $c$th channel after GCN operation. The features are aggregated from the features of all nodes in dimension $t$ of the GCN input data. And according to Eq (2-11), the features of GCN input data in dimension $t$ are determined by the $k$ numbers of input data before the moment $t$. Thus, we can obtain the relationship between the convolutional input and the GCN output as Eq (2-13).

$$X^{(t-k):t} \xrightarrow{conv,GCN} \dot{Y}_{c,q}^t \qquad (2-13)$$

This indicates that the model can theoretically handle the non-alignment of the spatial correlation described by Eq (2-3). It is worth mentioning that according to Eq (2-3), the data of node $i$ is only related to the data of nodes before and including $i$. This problem can be solved by constructing the adjacency matrix in a proper way.

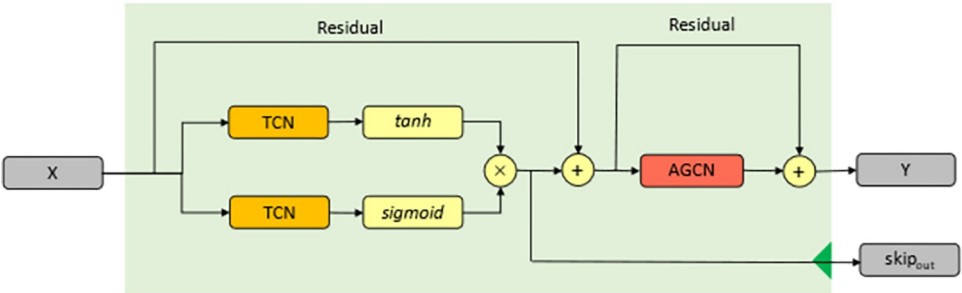

**Fig 7. Framework of a single spatio-temporal network (ST-Block).** Creating a spatio-temporal block by combining the WaveNet Network and Adaptive Graph Convolution Network previously described. Added residual network to AGCN.

The combination of a single convolution and a single GCN can handle data correlation up to $k$ time steps. To handle longer data association, there are two ways: one is to increase the value of $k$, and the other is to stack multiple modules. Stacking multiple modules not only handles longer data association, but also enhances the expressiveness of the model. Theoretically, after $r$ layers of iterations, the model can handle $rk$-length correlations, as shown in Eq (2-14).

$$X^{(t-rk):t} \underbrace{\xrightarrow{conv,GCN} \cdots \xrightarrow{conv,GCN}}_{r} \tilde{Y}_{c,q}^t \qquad (2-14)$$

This explains why the length of sequences that the model can handle grows linearly with the number of stacks. It is worth mentioning that the above analysis is performed in the context of ordinary convolution, while this paper uses dilated convolution. The length of the sequence that can be processed by dilated convolution increases exponentially with the number of stacks [29].

**Proposed model: W-WaveNet.** The model W-WaveNet fuses the WaveNet network, Adaptive graph convolution network (AGCN) network, and LSTM network mentioned in 2.2 by the fusion strategy above, and adds residual connections and skip connections.

The WaveNet network is first connected to the AGCN network, then the output of the TCN is drawn out as a skip connection, and finally the input and output of the AGCN network are connected together with a residual connection. the residual connection on the AGCN makes it easier to pass the gradient back to the TCN network when the network is trained. The network thus composed can handle temporal dependence as well as spatial dependence, and is therefore called a spatio-temporal network, whose structure is shown in Fig 7. For convenience, it will be referred to as ST-Block in the following.

According to the spatio-temporal network fusion strategy mentioned above, stacking multiple spatio-temporal networks can handle data association for longer periods of time. In order to prevent problems such as overfitting caused by too deep network layers, skip connections are drawn from each ST-Block to the end of all ST-Blocks, merged and then connected to the LSTM network, and finally the results are obtained through a fully connected layer. The final structure diagram is shown in Fig 8.

The ST-Block module in the upper layer is used to handle short-time spatial dependencies. The deeper the layer, the longer spatial dependencies can be handled by the ST-Block module.

## Algorithm

Describe the calculation process of the model with pseudo code:

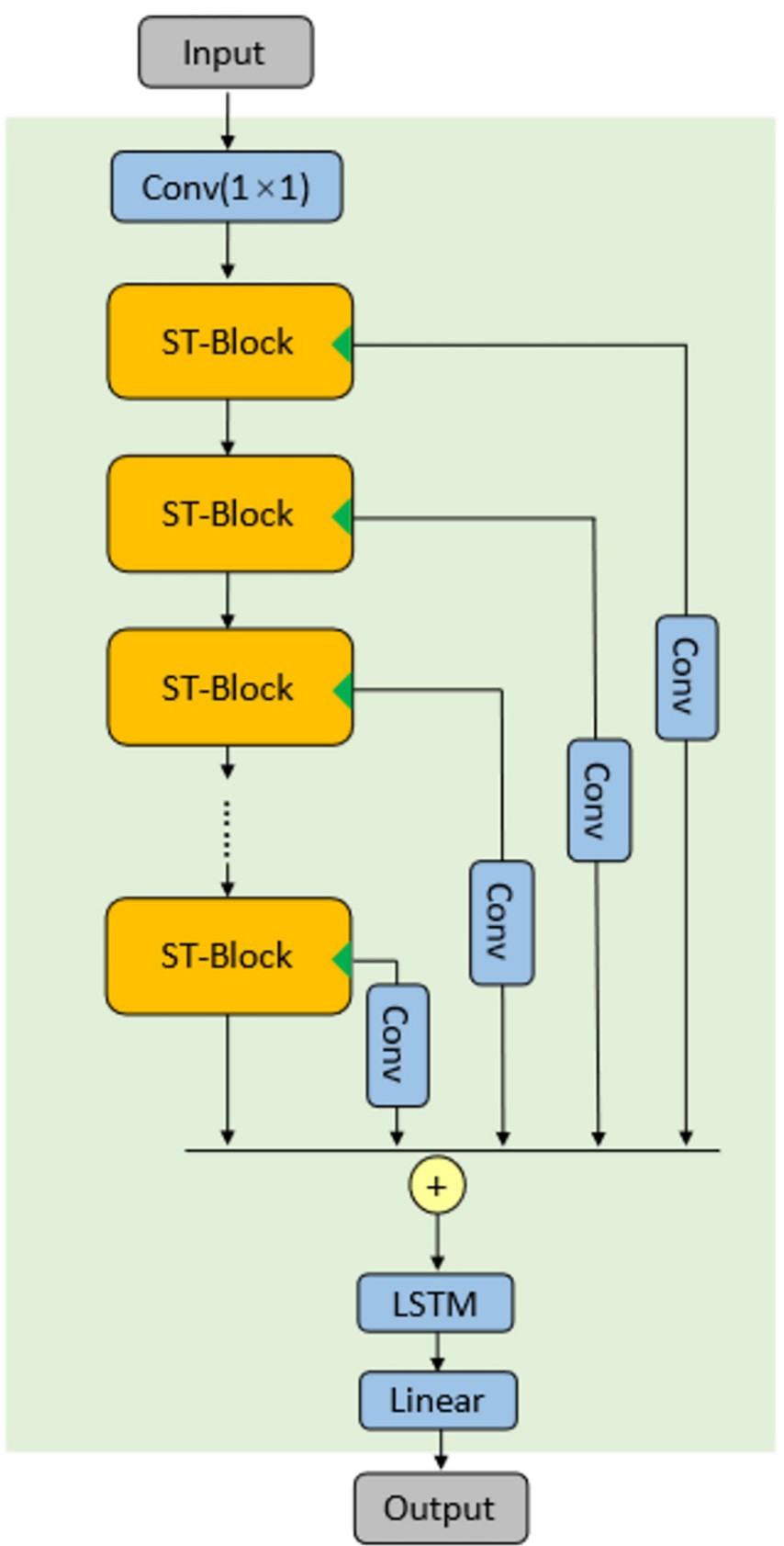

**Fig 8. Framework of W-WaveNet.** It combines the skip result of each spatio-temporal block with the stacked output, and finally compute the result through the LSTM network. All space-time blocks are connected to each other by skip connections.

```
Algorithm 1: training process
Input: training set {Xᵢ}ₙ, Xᵢ∈ℝⁿˣᴰ,n is number of nodes, D is length of
sequence
Output: spatial-temporal graph convolution model M(x)
For each epoch:
      For each batch:
            X_start←Conv(X_in) // dilated causal convolution
            C_α ← SoftMax(ReLU(E₁E₂ᵀ)) // calculate node embedding matrix
            Y_skip←0 // initial value
            For each spatial-temporal layers:
                  if first layer:
                        Xᵢ←X_start
                  else:
                        Xᵢ←Y₁ // last Y connected to current X
            X_w ← Convᵀ(Xᵢ) ⊗ Convᴳ(Xᵢ) // gated dilated causal
convolution
                  X_w←X_w+Xᵢ // residual connection
                  X_skip←Convˢ(X_w) // skip convolution
                  X_G←Gconv(X_w) // graph convolution
                  Y₁←X_G+X_w // residual connection
                  Y_skip←Y_skip+X_skip //
            Y_LSTM ← LSTM(Y_skip), Y_out ← DNN(Y_LSTM)
            Calculate MAE loss
            Update network parameters
```

The convolution neural network used in this paper is one-dimensional, and the time complexity of a single one-dimensional convolution layer is $O(m \cdot k \cdot c_{in} \cdot c_{out})$ Where $m$ represents the output characteristic length, $k$ denotes the convolution kernel length, $c_{in}$ and $c_{out}$ indicates the number of input channels and output channels, respectively. The time complexity of a single-layer LSTM network is $O(h(d+3+h))$, and the time complexity of an adaptive graph convolution module is $O(n^3+n^2Lc)$, where $n$ is the number of graph nodes, $L$ represents the characteristic length of nodes, and $c$ denotes the number of channels after convolution. Therefore, the total time complexity is: $O(m \cdot k \cdot c_{in} \cdot c_{out} + h(d + 3 + h) + n^3 + n^2Lc)$.

## Results and discussion

### Baseline

In order to verify the effectiveness of this model, we need to build several models for comparison. Firstly, two shallow machine learning models, random forest (RF) and support vector regression (SVR), are constructed in this paper. Then, stack multiple WaveNet units mentioned above to build a deep learning model in which the internal parameters of each WaveNet unit are different. This model is called the WaveNet model in this paper. Based on the WaveNet model, the LSTM module is added to build the WaveNet-LSTM model. Add the MGCN module to build the WaveNet-MGCN model. Adding the LSTM module and MGCN module at the same time is called the W-WaveNet model.

The SVR model has many adjustable parameters. In order to avoid the influence of improper selection of super parameters on the SVR model, the bag method is used to adjust the super parameters of the SVR algorithm in this paper. A Bagging regressor is an integrated learning method that trains multiple predictors on random sub datasets. Finally, the results of

multiple predictors are aggregated by voting, averaging, and other methods, which effectively reduces the variance of prediction error.

The RF and SVR models used in this paper are implemented by the scikit-learn software package [32].

RF and SVR are both shallow machine learning models. In recent years, some studies have used deep learning methods to fit water pollution data. As far as we know, the CNN-LSTM model has been studied more. This paper constructs the CNN-LSTM hybrid model used by Barzegar et al. [16].

In addition, in order to verify that the model proposed in this paper has better performance in aggregating station information, the predicted results of the model are compared with the results of STGCN [23]. STGCN is also a spatial-temporal graph neural network model, which uses two convolution layers and a graph convolution to form spatial-temporal blocks and stacks these spatial-temporal blocks to build a deep learning network.

## Experiment setup

**Pretreatment.** The water quality data used in this paper comes from the data collected by the water quality instrument in the monitoring micro station. The process includes, but is not limited to, water sample collection, chemical determination, and data transmission. As long as one of these processes is abnormal, it will lead to a deviation between the final data and the actual water sample, or a lack of data. In the water quality data used in this study, some data even falls outside the theoretical scope. For example, some pH values are above 14, and some $NH_3$ and TP solubility are negative. The direct input of these data into model training will seriously affect the performance of the model, so the preprocessing process cannot be ignored.

The quartile method is used to detect outliers in this paper. Detected outliers are labeled as missing data. Then, the missing data is filled in by linear interpolation.

The input of the model needs fixed-length data, which is sampled by a sliding time window. The length of the window is 27. The first 24 values are labeled as input data, and the last three values are marked as output data. These 3-output data are represented by WQ (t + 1), WQ (t + 2), and WQ (t + 3), respectively. In other words, the model will use the first 24 water pollution data points to predict the data of the last 3-time steps.

Standardizing data can eliminate dimensions, reduce the impact of data representation on model results, and avoid paying too much attention to dimensions with large variance. All data is standardized with a Z-score in this study.

The models constructed in this paper include deep learning models and shallow machine learning models. When fitting water quality data with a depth model, the dataset needs to be divided. In this study, the data processed above are divided into training set, validation set and test set, with a ratio of 7:1:2.

**Analysis of spatial correlation.** In order to verify whether the data among different stations are related, pearson correlation coefficient (r) is applied to measure the degree of correlation between stations. When coupling the correlation between station A and station B, calculate the correlation coefficient between the data of station A and data from last 24-time steps in station B, and finally take the average value as the correlation between station A and station B. Use the data of TN to calculate and draw the heatmap as shown in Fig 9A. The horizontal and vertical coordinates are the site numbers, and the values in the figure are the correlation coefficients multiplied by 100 for display purposes. note that the larger the site number, the more downstream the site is. The results revealed that there is indeed a high correlation between some of the sites, and the correlation can be used to improve the accuracy of the model.

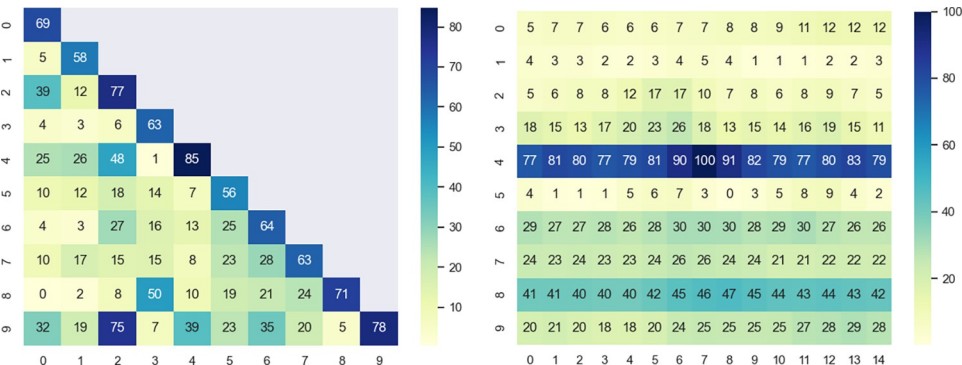

**Fig 9. Correlation heatmap of muti-site water quality data.** (A) Correlation heatmap among sites. The correlation between sites varies, some sites can be correlated up to 0.75. while some sites correlation is 0. (B) Correlation heatmap with site 4 time point 7 at different times and different site. The correlation between the same site and site 4 time point 7 reaches its maximum at a certain point, which is related to the distance between the site and site 4.

Because the sites are connected to each other through the river or pipe network, theoretically, the water quality pollution factor of an upstream site to affect the water quality pollution factor downstream, must wait until the upstream pollution factor spread to the downstream, and this time interval varies from site to site. Therefore, the correlation of data between different stations is not aligned by time. To demonstrate this, a heat map is plotted as shown in Fig 9B. The horizontal coordinates in the figure indicate the time and the vertical coordinates indicate the station number. Each value in the graph indicates the correlation coefficient between the corresponding time point of the corresponding station and station 4 at time point 7. Observing this heat map, it can be noticed that the data of site 4 at time point 7 do not all reach maximum correlation with the data of other sites at time point 7. Site 3 reaches maximum correlation at time point 26, site 5 reaches maximum correlation at time 12, and site 8 reaches maximum correlation at time 8. Obviously, each site does not have the same maximum relevance time point.

The model proposed in this paper aggregates data from multiple sites together for calculation, so the correlation of sites will have an impact on the model performance. If the sites are not correlated or the correlation is very small, the model will degrade to a single-site water quality prediction model. To measure the level of correlation of the data, we calculated the average number of correlated sites. When the correlation coefficient between two sites is greater than 0.25, we consider that there is a correlation between the two sites, and vice versa, we consider that there is no correlation between the sites. The number of sites associated with each site was calculated and finally averaged to obtain the average number of associated sites for the dataset. The final result is plotted as the bar chart in Fig 10.

Notice that the correlation of different pollutant factors varies between sites. It can be seen from the figure that the average number of correlated stations for the TP data of section A is only 0.5, which means that most of the stations are not correlated with each other and the model cannot improve the prediction effect on such dataset.

**Evaluation metrics.** Six measurement factors commonly used in water quality modeling were selected to assess the performance of the model. They are: mean absolute error ($MAE$) [33–35], mean absolute percentage error ($MAPE$) [36–39], root mean square error ($RMSE$) [36–39], root mean square percentage error ($RMSPE$) [35,40] coefficient of determination ($r2$) [17,40,41], and Pearson correlation coefficient ($r$) [33–35].

$MAE$ and $RMSE$ are used to describe the accuracy of the model. Their theoretical range is $-\infty$ to $+\infty$. The size of $MAE$ and $RMSE$ is related to the data dimension. Different water

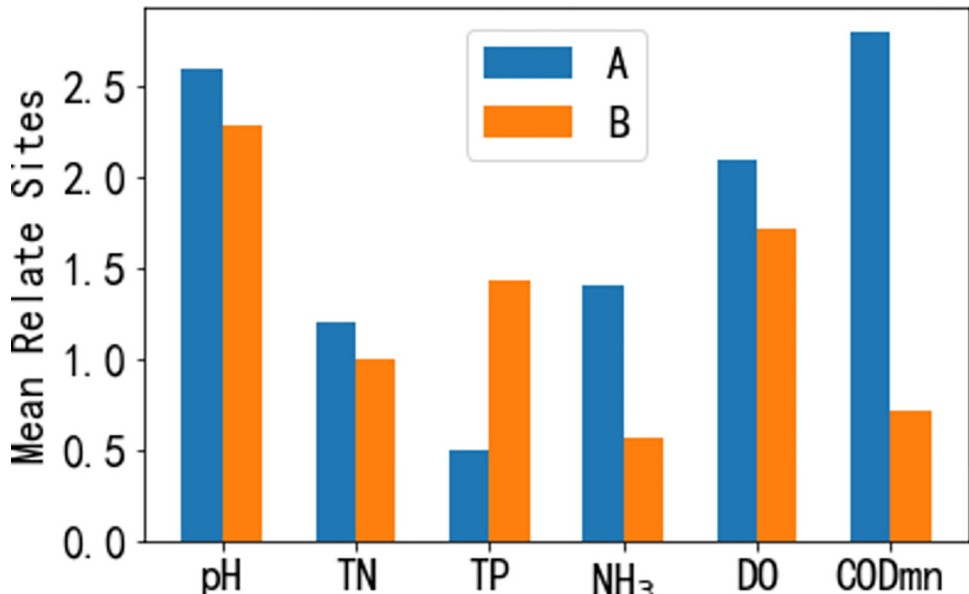

**Fig 10. Average number of correlated sites for each factor in Sections A and B.** This graph demonstrates the different site correlations that exist for the different factor data. Some of these data with low correlations may affect the results.

pollution factors have different statistical characteristics. Therefore, *MAPE* and *RMSPE* are used for horizontal comparison between factors, their theoretical range is 0 to $+\infty$. When *MAPE* or *RMSPE* is equal to 0, it means that the observed value is completely consistent with the predicted value. *r* is able to measure the linear correlation between the predicted value and the observed value. The theoretical range is from -1 to +1. When *r* is close to +1, it shows that the higher the linear correlation is, the better the prediction performance is. *r2* is the determination coefficient, the theoretical range is $-\infty$ to +1, and the ideal value is +1. *r2* is independent of the data dimension and can be used to compare the prediction effects between different models. When *r2* is greater than 0.75, the model is generally considered to have nice prediction performance.

## Discussion

In order to get the overall performance of each model, the average value of each measurement index of the model at all stations, with all pollution factors and 3-time steps is calculated. The overall performance of the model in sections A and B is shown in Tables 3 and 4. From the

**Table 3. The overall performance of each model in section A.**

| Models | MAE | MAPE | RMSE | RMSPE | r2 | r |
|---|---|---|---|---|---|---|
| RF | 0.381 | 0.113 | 0.627 | 0.091 | 0.873 | 0.935 |
| SVR | 0.375 | 0.116 | 0.630 | 0.091 | 0.869 | 0.934 |
| CNN-LSTM | 0.460 | 0.158 | 0.709 | 0.116 | 0.824 | 0.917 |
| STGCN | 0.302 | 0.092 | 0.494 | 0.072 | 0.912 | 0.956 |
| WaveNet | 0.367 | 0.103 | 0.640 | 0.088 | 0.868 | 0.933 |
| WaveNet-LSTM | 0.363 | 0.102 | 0.633 | 0.087 | 0.871 | 0.934 |
| WaveNet-MGCN | 0.326 | 0.094 | 0.569 | 0.081 | 0.894 | 0.946 |
| W-WaveNet | **0.225** | **0.070** | **0.406** | **0.061** | **0.939** | **0.969** |

**Table 4. The overall performance of each model in section B.**

| Models | MAE | MAPE | RMSE | RMSPE | r2 | r |
|---|---|---|---|---|---|---|
| RF | 0.406 | 0.116 | 0.663 | 0.094 | 0.867 | 0.931 |
| SVR | 0.400 | 0.120 | 0.667 | 0.094 | 0.863 | 0.930 |
| CNN-LSTM | 0.511 | 0.133 | 0.814 | 0.102 | 0.819 | 0.911 |
| STGCN | 0.406 | 0.114 | 0.624 | 0.086 | 0.889 | 0.943 |
| WaveNet | 0.392 | 0.106 | 0.674 | 0.091 | 0.863 | 0.930 |
| WaveNet-LSTM | 0.391 | 0.106 | 0.669 | 0.091 | 0.864 | 0.930 |
| WaveNet-MGCN | 0.378 | 0.102 | 0.645 | 0.088 | 0.872 | 0.935 |
| W-WaveNet | **0.287** | **0.083** | **0.504** | **0.073** | **0.912** | **0.955** |

table, the W-WaveNet model proposed in this paper has the best indicators in all aspects and has been greatly improved compared with other models. In addition to the model proposed in this paper, the best model is STGCN, common to them is that spatial correlation is considered. This indicates that modeling spatial relationships can improve model performance. Surprisingly, the CNN-LSTM model has a larger prediction error than the shallow machine learning models RF and SVR. The possible reason is that the hyperparameters do not match the current dataset, which also suggests that some current water quality pollution models may only be applicable to specific datasets. W-WaveNet achieves optimal performance, The MAE of section a is reduced from 0.460 to 0.240 of CNN-LSTM model, and that of section B is reduced from 0.511 to 0.287 of CNN-LSTM model, with an average reduction of 45.8%; r2 of section A is increased from 0.824 of CNN-LSTM model to 0.933, and r2 of section B is increased from 0.819 of CNN-LSTM model to 0.912, with an average increase of 12.3%. Compared with W-WaveNet, WaveNet-LSTM lacks the AGCN network. Its MAE, MAPE, and RMSE for predicting water quality at section A are 0.363, 0.102, and 0.633, respectively. The performance is only better than that of shallow machine learning. This indicates that the AGCN module in W-WaveNet plays a considerable role in reducing the model MAE by 38.0%. Compared with W-WaveNet, WaveNet-AGCN lacks an LSTM network. Its MAE, MAPE, and RMSE for predicting water quality at section a are 0.326, 0.094, and 0.569, respectively. The performance is also very poor compared with W-WaveNet, indicating that the LSTM at the end of the model reduces the MAE error of W-WaveNet by 30.9%.

To demonstrate the model training process, an error profile is plotted as shown in Fig 10. After each round of training, the generalization performance of the current model is calculated on the validation set. Deep learning models often have over-fitting problems. As a result, the early stop method is adopted in the process of model training. The early stop [42] method is a widely used method that has been proven to be better than the regularization method in many cases. Therefore, the total number of rounds when the model terminates training is not fixed. Due to space limitations, this paper only shows the training error and validation error curve of the W-WaveNet model under all factors of section A, as shown in Fig 11. From the figure, the training of the model on the five factors other than TP is in line with the expected effect, the training set error and validation set error decrease slowly, and the training set error is smaller than the validation set error. But in the case of TP, the error of the training set is always close to that of the validation set, which shows that the data fitting effect of TP is not good. As can be seen from Fig 9, the average number of site associations for TP is low, and it is speculated that this may due to the difficulty of training the model by forcing unrelated site data to be aggregated together. From the following table and plot, it can be seen that the model proposed in this paper does not have the best results in predicting TP.

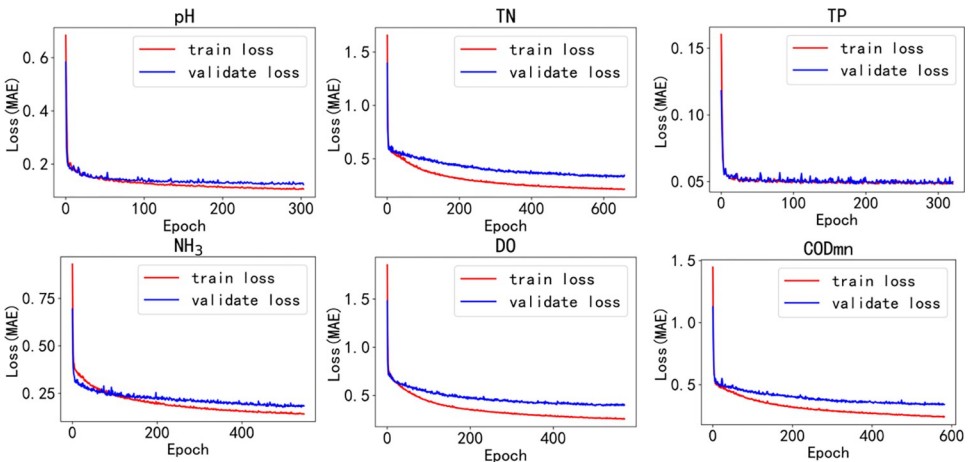

**Fig 11. Training error and validation error curves of W-WaveNet model under all factors of section A.**

Tables 3 and 4 show the average performance of the models under all factors. To verify whether the models achieve excellent prediction performance under each factor, Tables 5 and 6 are listed to show the MAE, RMSE, and r2 of the six models under all pollution factors in Section A. The smaller the *MAE* or *RMSE* value, the better the prediction performance, while the larger the *r2* value, the better the prediction effect. According to the table, the W-WaveNet model has the best prediction performance on pH value, $NH_3$, and TP. Because the average number of site associations for TP is very low, the prediction is not optimal, but it is similar to the best model. We also evaluated the model on data of DO and $COD_{mn}$, and again W-WaveNet achieved the best results, but for reasons of space, detailed results are not shown.

Taking $NH_3$ as an example, the MAE results of SVR, RF, CNN-LSTM, STGCN, and W-WaveNet are 0.274, 0.283, 0.354, 0.258, and 0.165, respectively, in predicting the first time step. The CNN-LSTM model has the worst prediction, and SVR and RF have similar results. The STGCN model aggregates multi-site information and reduces the prediction error by

**Table 5. MAE, RMSE and r2 for each model at pH and $NH_3$ in section A.**

| Time step | Models | pH | | | $NH_3$ | | |
|---|---|---|---|---|---|---|---|
| | | MAE | RMSE | r2 | MAE | RMSE | r2 |
| WQ(t+1) | SVR | 0.116 | 0.224 | 0.948 | 0.274 | 0.510 | 0.888 |
| | RF | 0.117 | 0.225 | 0.947 | 0.283 | 0.511 | 0.888 |
| | CNN-LSTM | 0.134 | 0.229 | 0.945 | 0.354 | 0.599 | 0.843 |
| | STGCN | 0.150 | 0.241 | 0.939 | 0.258 | 0.456 | 0.909 |
| | W-WaveNet | **0.087** | **0.176** | **0.967** | **0.165** | **0.322** | **0.955** |
| WQ(t+2) | SVR | 0.171 | 0.309 | 0.900 | 0.361 | 0.620 | 0.835 |
| | RF | 0.166 | 0.301 | 0.905 | 0.36 | 0.599 | 0.846 |
| | CNN-LSTM | 0.181 | 0.306 | 0.902 | 0.438 | 0.681 | 0.799 |
| | STGCN | 0.172 | 0.28 | 0.918 | 0.262 | 0.458 | 0.909 |
| | W-WaveNet | **0.108** | **0.217** | **0.951** | **0.175** | **0.321** | **0.955** |
| WQ(t+3) | SVR | 0.201 | 0.367 | 0.860 | 0.384 | 0.646 | 0.820 |
| | RF | 0.197 | 0.352 | 0.871 | 0.382 | 0.612 | 0.839 |
| | CNN-LSTM | 0.217 | 0.364 | 0.861 | 0.449 | 0.707 | 0.784 |
| | STGCN | 0.192 | 0.315 | 0.895 | 0.287 | 0.491 | 0.896 |
| | W-WaveNet | **0.120** | **0.237** | **0.941** | **0.187** | **0.337** | **0.951** |

**Table 6. MAE, RMSE and r2 for each model at TP and TN in section A.**

| Time step | Models | TP | | | TN | | |
|---|---|---|---|---|---|---|---|
| | | MAE | RMSE | r2 | MAE | RMSE | r2 |
| WQ(t+1) | SVR | 0.052 | 0.078 | 0.925 | 0.400 | 0.735 | 0.909 |
| | RF | 0.042 | 0.077 | 0.927 | 0.432 | 0.751 | 0.905 |
| | CNN-LSTM | 0.093 | 0.122 | 0.806 | 0.482 | 0.788 | 0.949 |
| | STGCN | 0.046 | 0.078 | 0.921 | 0.437 | 0.695 | 0.922 |
| | W-WaveNet | **0.036** | **0.070** | **0.937** | **0.296** | **0.541** | **0.953** |
| WQ(t+2) | SVR | 0.062 | 0.091 | 0.897 | 0.564 | 0.977 | 0.842 |
| | RF | 0.052 | **0.088** | **0.904** | 0.581 | 0.98 | 0.841 |
| | CNN-LSTM | 0.111 | 0.142 | 0.740 | 0.643 | 1.063 | 0.816 |
| | STGCN | 0.054 | 0.093 | 0.889 | 0.466 | 0.766 | 0.905 |
| | W-WaveNet | **0.048** | 0.091 | 0.894 | **0.318** | **0.568** | **0.948** |
| WQ(t+3) | SVR | 0.067 | 0.099 | 0.877 | 0.648 | 1.080 | 0.804 |
| | RF | 0.058 | **0.095** | **0.887** | 0.66 | 1.059 | 0.811 |
| | CNN-LSTM | 0.101 | 0.254 | 0.137 | 0.710 | 1.120 | 0.796 |
| | STGCN | 0.059 | 0.102 | 0.867 | 0.53 | 0.839 | 0.886 |
| | W-WaveNet | **0.054** | 0.102 | 0.867 | **0.355** | **0.610** | **0.939** |

8.8% compared to the RF model. The W-WaveNet model proposed in this paper has the best prediction performance, and the prediction error is reduced by 41.7% compared with the RF model. In predicting the second time step, the MAE of each model was 0.361, 0.360, 0.438, 0.262, and 0.175, respectively. In comparison to the RF model, the W-WaveNet model's result is decreased by 51.2 percent at this time step. The MAE of each model in forecasting the third time step was 0.384, 0.382, 0.449, 0.287, 0.187, respectively. At this time step, the W-WaveNet model is reduced by 51.3% compared to the RF model. This indicates that the W-WaveNet model has great stability. The error growth is not severe when predicting longer time steps. pH, TP, and TN are similar and will not be repeated.

In order to intuitively show the difference in prediction effects between different models, the histogram is present to show the MAPE and RMSPE results of each model, as shown in Fig 12. Limited to space, this paper only shows the model histogram results of section A. It can be concluded from the figure that the prediction effect of the W-WaveNet model is excellent, especially under TN, DO, and $NH_3$, which is much better than that of other models. Considering MAPE and RMSPE are independent of data measurement, horizontal factor comparison can be compared. The prediction performance of pH value is the best, while the prediction performance of DO is the worst, which is related to the characteristics of the data itself.

Scatter plots can visualize the linear relationship between the predicted and observed values. The scatter plots of predicted and observed values for each model when predicting the TN at section A on the test set are shown in Fig 13. The SVR, RF, and CNN-LSTM models all have more scattered scatter plots, indicating poor prediction. When compared to the previous three models, the scatter plots of the STGCN model are substantially more concentrated, suggesting that the STGCN model efficiently combines data across sites. In comparison to the STGCN model, the scatter plot of the W-WaveNet model is more focused and contains fewer points that stray far from the straight line. This implies that the W-WaveNet model better describes the correlation among multi-site water quality data.

In terms of the regression straight line, the slope and intercept for the SVR model are 0.859 and 0.860, respectively. The RF model has a slope of 0.841 and an intercept of 1.049. The CNN-LSTM model has a slope of 0.869 and an intercept of 0.947. The STGCN model has a

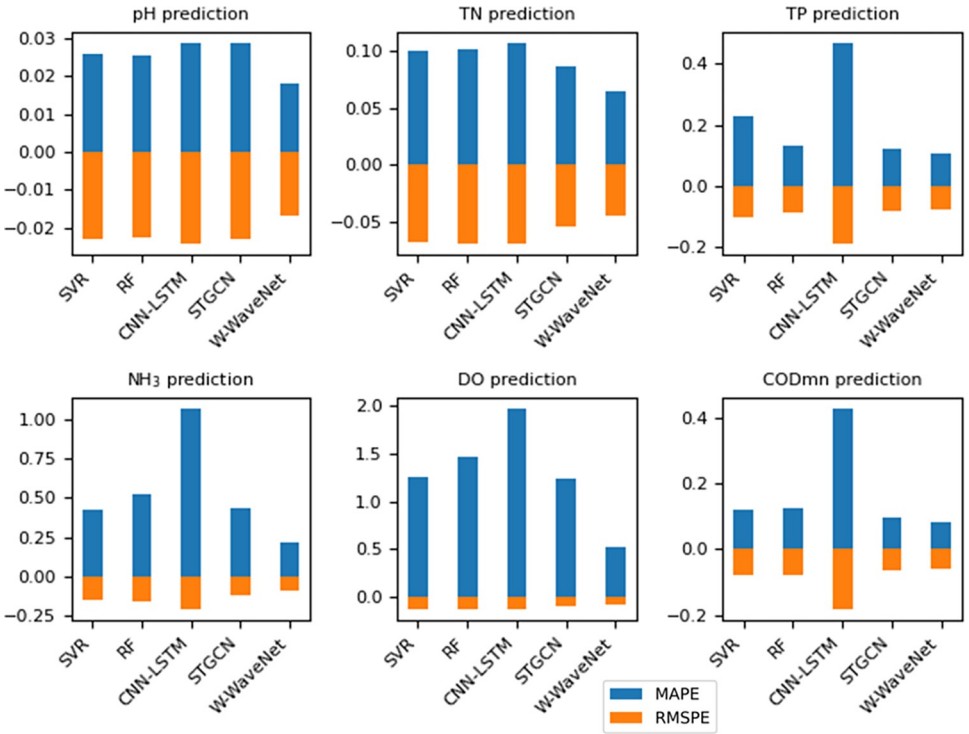

**Fig 12. Histogram of MAPE and RMSPE for each model under all factors of section A.** With all factor data, the figure demonstrates that W-WaveNet produces the best results or outcomes that are close to the best results.

slope of 0.921 and an intercept of 0.502. The W-WaveNet model has a slope of 0.956 and an intercept of 0.284. The W-WaveNet model is obviously closer to the line y = x than the other models and best fits the data.

Fig 14. depicts each model's time-series plot while forecasting data for the third time step of $NH_3$. The W-WaveNet model has the closest predicted values to the observed values in this figure, whereas the CNN-LSTM model performs the worst.

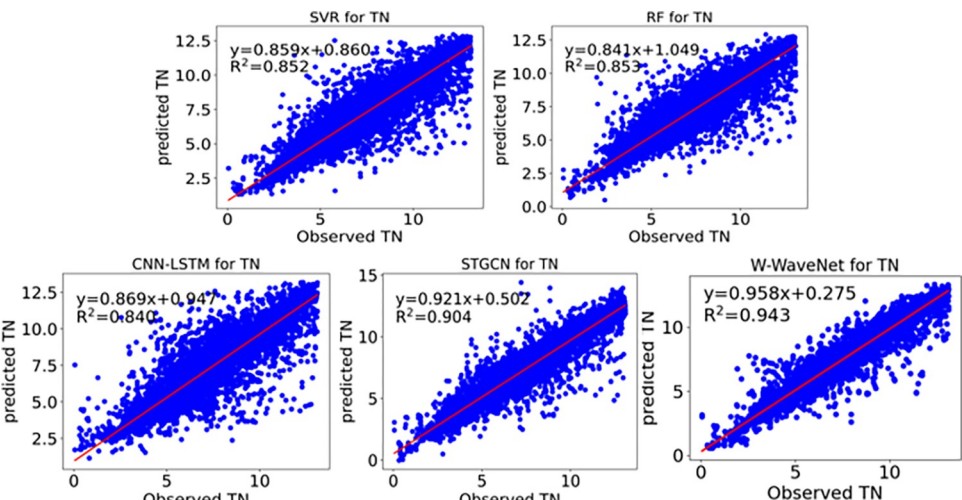

**Fig 13. Scatter plot of TN of each model in section A.** This plot shows that the predicted scatter plot of W-WaveNet is closer to the straight line compared to the other models.

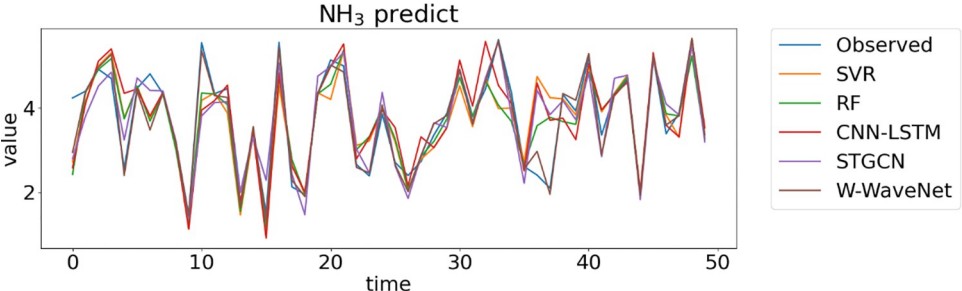

**Fig 14. Time-series plots of NH₃ at section A predicted by each model.** This plot shows that the prediction curve of W-WaveNet is closer to the observed curve than other models.

In summary, the model proposed in this paper performs well on the dataset with a high average number of correlated sites, and the prediction effect is greatly improved. However, the prediction results on the dataset with low average number of associated sites are similar to other models. This actually indicated that the model effectively used the association information among sites to make predictions.

In summary, the model proposed in this paper performed well on the dataset with high spatial correlation (pH, TN, NH₃, DO, and COD$_{mn}$), and the prediction effect was greatly improved. However, the prediction results for the less spatially correlated dataset (TP) were similar to the other models. This actually shows that the model effectively uses the spatial correlation information among sites for prediction. W-WaveNet is an end-to-end black-box model that only requires data cleaning, and the rest of the process is automatic. Moreover, the code of the model is publicly available, This means that the model can be easily applied to other datasets with spatio-temporal correlations, such as traffic, ecology, climate, etc. These data are ubiquitous worldwide.

## Conclusions

Existing water quality prediction models don't take into account spatial correlations among sites, so this study combines the WaveNet network, LSTM network, and adaptive graph convolutional network to create a W-WaveNet network that can handle not only spatial correlations but also complex time-dependent relationships.

The spatial correlations among stations have different degrees of time lags due to factors such as inter-site distance and water velocity, which are referred to as non-aligned spatial correlations in this paper. Theoretical analysis shows that interleaved stacking of spatio-temporal networks can deal with this problem. Stacking multiple networks also allows the model to handle both short-term and long-term dependence. Adding multiple residual connections and skip connections to the model makes it easier to converge.

In the experimental part, this study validated the performance of the model on two real water quality cross-sections, each dataset using the average number of associated sites to measure the degree of spatial association. The experimental results of the model were compared with the results of SVR, RF, CNN-LSTM, and STGCN using a variety of measurement factors. The results show that the model proposed in this study can substantially improve the prediction in cases where the correlation among sites does exist.

The model proposed in this research could be better because it ignores the correlation among pollution factors. We will include this factor in our future work. Besides, we will enhance the spatial interpretability of the model and verify the validity of the model using spatio-temporal data in other domains.

## Author Contributions

**Conceptualization:** Shangping Zhong, Kaizhi Chen.

**Data curation:** Shangping Zhong, Kaizhi Chen.

**Formal analysis:** Shaojun Yang.

**Investigation:** Shaojun Yang, Kaizhi Chen.

**Methodology:** Shaojun Yang.

**Project administration:** Shaojun Yang.

**Resources:** Shangping Zhong, Kaizhi Chen.

**Software:** Shaojun Yang.

**Supervision:** Shaojun Yang, Shangping Zhong.

**Validation:** Shaojun Yang, Shangping Zhong.

**Visualization:** Shaojun Yang.

**Writing – original draft:** Shaojun Yang.

**Writing – review & editing:** Shangping Zhong, Kaizhi Chen.

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
