## [Decision Letter · Decision Letter 0]

8 Aug 2022

PONE-D-22-18978W-WaveNet: A multi-site water quality prediction model incorporating adaptive graph convolution and CNN-LSTMPLOS ONE

Dear Dr. yang,

Thank you for submitting your manuscript to PLOS ONE. After careful consideration, we feel that it has merit but does not fully meet PLOS ONE’s publication criteria as it currently stands. Therefore, we invite you to submit a revised version of the manuscript that addresses the points raised during the review process. There is no requirement from the journal to cite these specific papers, unless you deem that they are genuinely necessary in order to provide context to the study

We look forward to receiving your revised manuscript.

Kind regards,

Sathishkumar V E

Academic Editor

PLOS ONE

Journal Requirements:

2. Please ensure that all data sources used are listed in both the Methods section and the Data availability statement in the submission form, including locations from where real data was collected, and how this was obtained by the authors.

3. Please amend your Data availability statement to declare where data can be found. Please explain any restrictions on data sharing, and please note that having an author contact for data availability is not acceptable.

4. In your Methods section, please provide additional information regarding the permits you obtained for the work. Please ensure you have included the full name of the authority that approved the field site access and, if no permits were required, a brief statement explaining why. Please also include geographical coordinates or location information for samples collected in the field, if available.

Reviewers' comments:

Reviewer's Responses to Questions

**Comments to the Author**

1. Is the manuscript technically sound, and do the data support the conclusions?

Reviewer #1: No

Reviewer #2: Yes

Reviewer #3: Yes

2. Has the statistical analysis been performed appropriately and rigorously? 

Reviewer #1: No

Reviewer #2: Yes

Reviewer #3: Yes

3. Have the authors made all data underlying the findings in their manuscript fully available?

Reviewer #1: Yes

Reviewer #2: Yes

Reviewer #3: Yes

4. Is the manuscript presented in an intelligible fashion and written in standard English?

Reviewer #1: No

Reviewer #2: Yes

Reviewer #3: Yes

5. Review Comments to the Author

Reviewer #1: The Research Paper stands Rejected and is NOT RECOMMENDED for Publication because of the following reasons:

1. The Conceptual methodology and outline of the manuscript has weak analysis and even little stress on the proposed methodology.

2. Literature review, and even some other sections like System Model, Analysis is missing.

3. Experimental Results are little bit confusing and not organized properly.

4. The Language of the paper has not suitable flow.

5. Overall the paper has weak methodology and quality is not appreciated.

Reviewer #2: The paper W-WaveNet: A multi-site water quality prediction model incorporating adaptive graph

convolution and CNN-LSTM speaks an important issue. The authors can strengthen the literature with following papers in their related works

-Saranya, A., Kottursamy, K., AlZubi, A.A. and Bashir, A.K., 2021. Analyzing fibrous tissue pattern in fibrous dysplasia bone images using deep R-CNN networks for segmentation. Soft Computing, pp.1-15.

-Arirangan, S. and Kottursamy, K., 2021. Multi‐scaled feature fusion enabled convolutional neural network for predicting fibrous dysplasia bone disorder. Expert Systems, p.e12882.

-L. Huang, R. Nan, K. Chi, Q. Hua, K. Yu, N. Kumar, and M. Guizani, "Throughput Guarantees for Multi-Cell Wireless Powered Communication Networks with Non-Orthogonal Multiple Access," IEEE Transactions on Vehicular Technology, 2022, doi: 10.1109/TVT.2022.3189699.

Y. Peng, A. Jolfaei and K. Yu, "A Novel Real-Time Deterministic Scheduling Mechanism in Industrial Cyber-Physical Systems for Energy Internet," IEEE Transactions on Industrial Informatics, vol. 18, no. 8, pp. 5670-5680, Aug. 2022, doi: 10.1109/TII.2021.3139357.

D. Xu, K. Yu and J. A. Ritcey, "Cross-Layer Device Authentication With Quantum Encryption for 5G Enabled IIoT in Industry 4.0," IEEE Transactions on Industrial Informatics, vol. 18, no. 9, pp. 6368-6378, Sept. 2022, doi: 10.1109/TII.2021.3130163.

Y. He, L. Nie, T. Guo, K. Kaur, M. M. Hassan, and K. Yu," A NOMA-Enabled Framework for Relay Deployment and Network Optimization in Double-Layer Airborne Access VANETs," IEEE Transactions on Intelligent Transportation Systems, doi: 10.1109/TITS.2021.3139888.

Y. Lu, L. Yang, S. X. Yang, Q. Hua, A. K. Sangaiah, T. Guo, and K. Yu, “An Intelligent Deterministic Scheduling Method for Ultra-Low Latency Communication in Edge Enabled Industrial Internet of Things,” IEEE Transactions on Industrial Informatics, 2022, doi: 10.1109/TII.2022.3186891.

J. Wei, Q. Zhu, Q. Li, L. Nie, Z. Shen, K. -K. R. Choo, K. Yu, “A Redactable Blockchain Framework for Secure Federated Learning in Industrial Internet-of-Things”, IEEE Internet of Things Journal, doi: 10.1109/JIOT.2022.3162499.

Reviewer #3: The topic of the manuscript ID: PONE-D-22-18978 is interesting, promising, and within the scope of the journal. However, before the article is accepted for publication, the authors should address the following comments:

1. Authors need to improve the introduction section by highlighting major difficulties and challenges, and your original achievements to overcome them.

2. Why did the authors choose two sections of a river basin in Fujian as a case study?

3. Write the full form of all the used abbreviations as they come in the paper.

4. Include a clear map of the study location along with sites.

5. Write the unit of the water quality parameters.

6. Improve the quality of all Figures, and also write their captions.

7. What are the advantages of the applied models over others in modelling complex hydrological processes? Explain

8. Did the training data undergo any pre-processing?

9. Many studies have already proved the potential of machine learning/deep learning models in water quality prediction. What is the transferability of such results to other locations in terms of impact or usefulness? Is it really the novelty in a true and specific sense or just to test the applied models?

10. Why did the authors choose six evaluation metrics i.e., Mean Absolute Error (MAE), Mean Absolute Percentage Error (MAPE), Root Mean Square Error (RMSE), Root Mean Square Percentage Error (RMSPE), coefficient of determination (r2), and Pearson correlation coefficient (r)? Justify, also write their range and cite all these by citing the appropriate references.

11. Authors need to furnish more discussion about the practical utility of work in the discussion section and its usefulness on a global scale.

12. In conclusion, it includes the direction for future works.

13. The reviewers recommend some useful references, that need to be cited which help the authors for improvement of the paper.

Modelling of Bunus regional sewage treatment plant using machine learning approaches. Desalination and Water Treatment, 203: 80-90, https://doi.org/10.5004/dwt.2020.26160.

Effluent’s quality prediction by using nonlinear dynamic block-oriented models: a system identification approach. Desalination and Water Treatment, 218: 62-52, https://doi.org/10.5004/dwt.2021.26983.

Integrating feature extraction approaches with hybrid emotional neural networks for water quality index modeling. Applied Soft Computing, https://doi.org/10.1016/j.asoc.2021.108036.

Comparative implementation between neuro-emotional genetic algorithm and novel ensemble computing techniques for modelling dissolved oxygen concentration. Hydrological Sciences Journal, 66(10): 1584-1596, https://doi.org/10.1080/02626667.2021.1937179.

6. PLOS authors have the option to publish the peer review history of their article (what does this mean?). If published, this will include your full peer review and any attached files.

Reviewer #1: **Yes: **Anand Nayyar

Reviewer #2: No

Reviewer #3: No

---

## [Author Response · Author response to Decision Letter 0]

27 Sep 2022

To Prof. Sathishkumar V E:

We have revised the manuscript according to the formatting requirements, if there are still omissions, please feel free to point them out. As for the dataset, we decided to make it desensitized and public, and users can only apply it for teaching and research, not for commercial use. We have also added more information about the dataset in the manuscript.

The following is the responses to reviewers.

To Reviewer #1:

Thank you very much for your criticism, we have carefully checked our article and revised it in response to your comments. Your criticism has made our article better and we would be very appreciative of more specific comments. The following is the reply point by point.

Point 1:

Referee: The Conceptual methodology and outline of the manuscript has weak analysis and even little stress on the proposed methodology.

Reply:

The deep learning used in the article is a black-box model, and there is no advanced conceptual methodology. We have incorporated several models for the challenges in multi-site water quality problems. Each model is analyzed. The innovation of the article is the fusion of existing models and the adaptability of the fusion approach to the multi-site water quality problem. We also analyze theoretically the process of model fusion approach to spatio-temporal dependence.

Point 2:

Referee: Literature review, and even some other sections like System Model, Analysis is missing.

Reply:

We have reorganized the literature review section and highlighted the analysis section.

The model proposed in the article consists of several fused components. We introduce and analyze each part of the model in the section "Relative models". The fusion method is analyzed in the section "Spatio-temporal network fusion strategy".

We have added some analysis to the "Discussion" section.

Point 3:

Referee: Experimental Results are little bit confusing and not organized properly.

Reply:

We have added some descriptive information and analysis to make the Experimental Results more understandable.

Point 4:

Referee: The Language of the paper has not suitable flow.

Reply:

We are very sorry that English is not our first language, but the article has been polished.

Point 5:

Referee: Overall the paper has weak methodology and quality is not appreciated.

Reply:

We have added some analysis to the article.

To the best of our knowledge, this paper is the first to delve into the problem of modeling multi-site water quality pollution. Our model is an end-to-end deep learning model and with open source code and dataset, it can be easily extended to other problems with spatio-temporal dependencies. We believe our study is valuable.

To Reviewer #2:

We appreciate your affirmation and the papers you provided. We have read these papers carefully, some of them were indeed helpful to the article, and we have cited them.

To Reviewer #3:

Thank you very much for your review and affirmation. Your comments are very professional and helpful to improve our article. The following is the reply point by point.

Point 1:

Referee: Authors need to improve the introduction section by highlighting major difficulties and challenges, and your original achievements to overcome them.

Reply:

Thank you for your reminder, we have written major difficulties and our original achievements in the summary part of introduction.

Point 2:

Referee: Why did the authors choose two sections of a river basin in Fujian as a case study?

Reply:

High quality multi-site data is not easy to obtain, and we were able to access these two cross-sections only after working with the Department of Environmental Protection.

Point 3:

Referee: Write the full form of all the used abbreviations as they come in the paper.

Reply:

We have added the full names of all abbreviations. If there are still omissions, feel free to point them out.

Point 4:

Referee: Include a clear map of the study location along with sites.

Reply:

We have added a map to the "Dataset describtion" and marked the location of the sites.

Point 5:

Referee: Write the unit of the water quality parameters.

Reply:

We have added unit of the water quality parameters in chapter “dataset describtion”.

Point 6:

Referee: Improve the quality of all Figures, and also write their captions.

Reply:

Our images are in high-quality png format and were converted to tif format using the conversion tool provided by the journal. Download the image source files to see the high-quality images. We have written more detailed captions for some of the images. Note that the captions are in the main body.

Point 7:

Referee: What are the advantages of the applied models over others in modelling complex hydrological processes? Explain

Reply:

Compared to other models, the greatest advantage of our proposed model is that it models complex spatial dependencies, allowing the model to handle water quality dependencies among multiple neighboring sites. We have added this description in the introduction.

Point 8:

Referee: Did the training data undergo any pre-processing?

Reply:

Yes, we describe the pre-processing process in the chapter "Pretreatment." Processing includes anomaly detection, data interpolation, sampling, and normalization.

Point 9:

Referee: Many studies have already proved the potential of machine learning/deep learning models in water quality prediction. What is the transferability of such results to other locations in terms of impact or usefulness? Is it really the novelty in a true and specific sense or just to test the applied models?

Reply:

Essentially, our work is modeling of spatio-temporal data. Since the model is an end-to-end deep learning model, it can be easily applied to other domains of spatiotemporal data. In our future work, we will try to apply the model to other spatio-temporal data. Meanwhile, we open the source code and welcome other researchers to try the performance of the model in other domains.

We believe that this research is novel. To the best of our knowledge, our article is the first to study the spatial correlation of multi-site water quality data. Although the model used in the article is not novel. However, we describe the applicability of the model to spatio-temporal water quality data in the section "Spatio-temporal network fusion strategy". We believe that this work is novel and meaningful.

Point 10:

Referee: Why did the authors choose six evaluation metrics i.e., Mean Absolute Error (MAE), Mean Absolute Percentage Error (MAPE), Root Mean Square Error (RMSE), Root Mean Square Percentage Error (RMSPE), coefficient of determination (r2), and Pearson correlation coefficient (r)? Justify, also write their range and cite all these by citing the appropriate references.

Reply:

We chose these six evaluation factors because, according to our survey, they are more frequently used in water quality model evaluation and can assess the effectiveness of the model from different perspectives. They are described in detail in the "evaluation metrics" section.

Point 11:

Referee: Authors need to furnish more discussion about the practical utility of work in the discussion section and its usefulness on a global scale. 

Reply:

We have added the usefulness of the model on a global scale and the practical utility at the end of the discussion.

Point 12:

Referee: In conclusion, it includes the direction for future works.

Reply:

Thanks to your reminder, we have added future work in the end of the conclusion.

Point 13:

Referee: The reviewers recommend some useful references, that need to be cited which help the authors for improvement of the paper.

Reply:

Thank you very much for the recommended paper, it is very relevant to our work and helps a lot.

---

## [Decision Letter · Decision Letter 1]

3 Oct 2022

W-WaveNet: A multi-site water quality prediction model incorporating adaptive graph convolution and CNN-LSTM

PONE-D-22-18978R1

Dear Dr. yang,

We’re pleased to inform you that your manuscript has been judged scientifically suitable for publication and will be formally accepted for publication once it meets all outstanding technical requirements.

Kind regards,

Sathishkumar V E

Academic Editor

PLOS ONE

Additional Editor Comments (optional):

Reviewers' comments:

Reviewer's Responses to Questions

**Comments to the Author**

1. If the authors have adequately addressed your comments raised in a previous round of review and you feel that this manuscript is now acceptable for publication, you may indicate that here to bypass the “Comments to the Author” section, enter your conflict of interest statement in the “Confidential to Editor” section, and submit your "Accept" recommendation.

Reviewer #1: All comments have been addressed

Reviewer #2: All comments have been addressed

2. Is the manuscript technically sound, and do the data support the conclusions?

Reviewer #1: Yes

Reviewer #2: Yes

3. Has the statistical analysis been performed appropriately and rigorously? 

Reviewer #1: Yes

Reviewer #2: Yes

4. Have the authors made all data underlying the findings in their manuscript fully available?

Reviewer #1: Yes

Reviewer #2: Yes

5. Is the manuscript presented in an intelligible fashion and written in standard English?

Reviewer #1: Yes

Reviewer #2: (No Response)

6. Review Comments to the Author

Reviewer #1: The Revised Paper has incorporated all the revisions as mentioned in the last review. Now the paper stands Accepted with no further revisions.

Reviewer #2: The authors addressed all the comments and Suggestions and now the paper is ready for acceptance.

7. PLOS authors have the option to publish the peer review history of their article (what does this mean?). If published, this will include your full peer review and any attached files.

Reviewer #1: No

Reviewer #2: No

---

## [Editor Report · Acceptance letter]

5 Oct 2022

PONE-D-22-18978R1 

W-WaveNet: A multi-site water quality prediction model incorporating adaptive graph convolution and CNN-LSTM 

Dear Dr. Yang:

I'm pleased to inform you that your manuscript has been deemed suitable for publication in PLOS ONE. Congratulations! Your manuscript is now with our production department. 

Kind regards, 

on behalf of

Dr. Sathishkumar V E 

Academic Editor

PLOS ONE